# Role of the *osaA* Gene in *Aspergillus fumigatus* Development, Secondary Metabolism and Virulence

**DOI:** 10.3390/jof10020103

**Published:** 2024-01-26

**Authors:** Apoorva Dabholkar, Sandesh Pandit, Ritu Devkota, Sourabh Dhingra, Sophie Lorber, Olivier Puel, Ana M. Calvo

**Affiliations:** 1Department of Biological Sciences, Northern Illinois University, DeKalb, IL 60115, USA; z1884764@students.niu.edu (A.D.); sha.sandesh@gmail.com (S.P.); 2Department of Biological Sciences and Eukaryotic Pathogen Innovation Center, Clemson University, Clemson, SC 29634, USA; rdevkot@g.clemson.edu (R.D.); sdhingr@clemson.edu (S.D.); 3Toxalim (Research Centre in Food Toxicology), Université de Toulouse, INRAE, ENVT, INP-Purpan, UPS, 31027 Toulouse, France; sophie.lorber@inrae.fr (S.L.); olivier.puel@inrae.fr (O.P.)

**Keywords:** *Aspergillus fumigatus*, OsaA, aspergillosis, WOPR domain, conidiation, secondary metabolism, virulence

## Abstract

*Aspergillus fumigatus* is the leading cause of aspergillosis, associated with high mortality rates, particularly in immunocompromised individuals. In search of novel genetic targets against aspergillosis, we studied the WOPR transcription factor OsaA. The deletion of the *osaA* gene resulted in colony growth reduction. Conidiation is also influenced by *osaA*; both *osaA* deletion and overexpression resulted in a decrease in spore production. Wild-type expression levels of *osaA* are necessary for the expression of the conidiation regulatory genes *brlA*, *abaA*, and *wetA*. In addition, *osaA* is necessary for normal cell wall integrity. Furthermore, the deletion of *osaA* resulted in a reduction in the ability of *A. fumigatus* to adhere to surfaces, decreased thermotolerance, as well as increased sensitivity to oxidative stress. Metabolomics analysis indicated that *osaA* deletion or overexpression led to alterations in the production of multiple secondary metabolites, including gliotoxin. This was accompanied by changes in the expression of genes in the corresponding secondary metabolite gene clusters. These effects could be, at least in part, due to the observed reduction in the expression levels of the *veA* and *laeA* global regulators when the *osaA* locus was altered. Importantly, our study shows that *osaA* is indispensable for virulence in both neutropenic and corticosteroid-immunosuppressed mouse models.

## 1. Introduction

The ubiquitous fungus *Aspergillus fumigatus* is one of the most common fungal pathogens of humans worldwide [1], particularly affecting immunosuppressed patients, including those with hematological malignancies (such as leukemia), solid-organ and hematopoietic stem cell transplant patients, individuals with prolonged corticosteroid therapy, patients with genetic immunodeficiencies such as chronic granulomatous disease, cancer patients undergoing chemotherapy, and individuals infected with HIV [2,3,4,5,6,7]. The main reason for the increase in systemic infections lies in the constant increase in the number of immunocompromised individuals [8], such as a higher number of transplants for end-organ disease, the development of immunosuppressive and myeloablative therapies for autoimmune and neoplastic diseases, the HIV pandemic [2,8,9,10], and the COVID-19 pandemic [11,12,13,14]. Mortality rates range from 40% to 90% [8,9,15,16]. The World Health Organization recently ranked *A. fumigatus* third in the critical priority group of fungal pathogens [17]. The leap from being an organism on the watch list in 2019 to a globally critical fungal pathogen in three years can be traced back to unmet health problems that are associated with aspergillosis [17,18].

For most patients, the main site of entry and infection for *A. fumigatus* is the respiratory tract. In immunosuppressed patients, *A. fumigatus* infection can cause aspergilloma and invasive aspergillosis (IA). Diagnostic methods are still limited, as well as the possibilities of therapeutic treatments, leading to high mortality rates [19,20,21]. *A. fumigatus* causes approximately 90% of systemic *Aspergillus* infections [22]. Fungal asexual spores (conidia) are only 2.5–3.0 μm in diameter, which allows them to reach the lung alveoli [23,24]. In healthy individuals, conidia that are not removed via mucociliary clearance encounter epithelial cells or alveolar macrophages, which are responsible for the phagocytosis and killing of conidia, as well as for initiating a pro-inflammatory response that recruits neutrophils (PMN type). IA is primarily a consequence of a dysfunction in these defenses. The most common host immunodeficiencies responsible for an increased risk of IA are neutropenia and corticosteroid-induced immunosuppression [2,25,26,27,28]. In neutropenic patients or in a murine model of chemotherapy-induced neutropenia, IA can cause thrombosis and hemorrhage due to rapid hyphal angioinvasion and systemic dissemination to other organs [9,29]. Corticosteroid-induced immunosuppressed patients usually present tissue necrosis and excessive inflammation. Corticosteroid treatments impair the function of phagocytes [30], thus rendering them unable to kill both conidia and hyphae [31,32,33,34,35,36]. In most cases, consequent excessive inflammation leading to tissue damage is considered the cause of death.

*Aspergillus fumigatus* possesses a variety of traits that allow it to become an opportunistic pathogen in the case of immunocompromised patients. Examples of these factors include thermotolerance and cell wall components of conidia [37,38,39,40,41]. Metabolic requirements can influence the propagation in vivo [42,43,44,45]. Secondary metabolites are considered part of the chemical arsenal required for niche specialization [46], including host–fungus interactions. *A. fumigatus* produces several toxic secondary metabolites, such as fumitoxin, fumigaclavines, fumigatin, fumagillin, verruculogen, fumitremorgins, gliotoxin, trypacidin, and helvolic acid [16,47,48,49]. Some of these mycotoxins act as immunosuppressants, which may be associated with pathogenesis processes.

Common treatments with azole drugs present side effects such as skin rashes and photopsia, especially in the early stages of treatment [50]. Studies have also revealed the presence of resistant isolates of *A. fumigatus* to azole treatment [51]. Prolonged azole therapy could result in the development of azole-resistant *A. fumigatus* strains during infection [52]. The widespread application of azole fungicides in crop fields is also a driving force for azole-resistant infections worldwide [53,54]. It is paramount to find new strategies against fungal infections such as aspergillosis, including novel genetic targets for the treatment or prevention of *A. fumigatus* infection. In the present study, we investigated the role of OsaA, a WOPR-domain transcription factor, in *A. fumigatus*.

The name WOPR evolved from closely related proteins, Wor1 (in *Candida albicans*), Pac2 (in *Schizosaccharomyce pombe*), and Ryp1 (in *Histoplasma capsulatum*), where these proteins play a major role in morphological development and pathogenesis [55,56,57]. Wor1 regulates white-opaque switching in *C. albicans* [55], affecting mating frequency and tissue specificity during infection [58]. Pac2 may mediate signaling between nutritional conditions and the activation of the transcription factor gene *ste11*, which is necessary for sexual development in *S. pombe* [56]. Ryp1, from the human pathogen *H. capsulatum*, is a regulator of the yeast–mycelia transition and affects the transcription of hundreds of genes [57]. In *Fusarium graminearum*, the WOPR protein Fgp1 is a key regulator of reproduction, toxin biosynthesis, and pathogenesis [59], as well as Sge1 in *Fusarium oxysporum* [60]. In *Aspergillus nidulans*, *osaA* was shown to be epistatic to the *veA* regulatory gene in maintaining a balance between sexual and asexual development [61].

Our present work revealed that *osaA* is necessary for proper colony growth and conidiation in *A. fumigatus*. In addition, our analysis indicated that secondary metabolite production was also influenced by *osaA*. Furthermore, *osaA* is indispensable for virulence in both the neutropenic and corticosteroid-immunosuppressed mouse models. Other virulence factors, such as cell wall stability, thermotolerance, adhesion capacity, and sensitivity to oxidative stress, were also decreased in the absence of *osaA* in this opportunistic human pathogen.

## 2. Materials and Methods

### 2.1. Sequence Analysis

The *Aspergillus fumigatus osaA* nucleotide and deduced amino acid sequence were acquired from FungiDB (https://fungidb.org/fungidb/app, 11 August 2020) with the accession number AFUB_093810. A search of OsaA homologs was carried out using the BLASTP search tool (https://blast.ncbi.nlm.nih.gov/, 11 June 2023). Clustal W (https://www.ebi.ac.uk/Tools/msa/clustalo/, 11 June 2023) and ESPript (https://espript.ibcp.fr/ESPript/ESPript/, 11 June 2023) were used for multiple sequence alignment (MSA). The phylogenetic tree was constructed using MEGA v11.0 software and a maximum likelihood model with a bootstrap value of 1000 (https://www.megasoftware.net/, 11 June 2023).

### 2.2. Strains and Culture Conditions

The strains of *A. fumigatus* used in this study are listed in Table 1. All the strains were grown in glucose minimal medium (GMM) [62] plus the appropriate supplements for the corresponding auxotrophic markers [62] unless otherwise indicated. A solid medium was prepared by adding 1% agar. Strains were stored as 30% glycerol stocks at −80 °C.

### 2.3. Generation of the osaA Deletion Strain

To construct the *osaA* deletion strain (Δ*osaA*) in *A. fumigatus*, a deletion of the DNA cassette was generated to replace the *osaA* gene with the *Aspergillus parasiticus pyrG* selectable marker [63]. First, the 5′UTR and 3′UTR were PCR-amplified from *A. fumigatus* genomic DNA using the primers osaA_F1, osaA_R2, and osaA_F3, osaA_R4, respectively (the primers used in this study are included in Appendix A). The *pyrG* marker gene was amplified from *A. parasiticus* genomic DNA using osaA_F5 and osaA_R6 primers. The three fragments were fused using the nested primers, osaA_F7, and osaA_R8, resulting in a 5157 bp PCR product. The obtained deletion fusion cassette was transformed in *A. fumigatus* CEA17 via polythene glycol-mediated fungal transformation. Transformants were selected on a stabilized minimal medium (SMM) [64] containing 1 M sucrose and no supplemental uracil. The Δ*osaA* strain was confirmed using a diagnostic PCR with primers osaA_F1 and A para pyrG_R and designated as TAD1.1.

### 2.4. Generation of the osaA Complementation Strain

To generate the *osaA* complementation strain (Com), a recombinant plasmid was constructed with pBC-hygro as the backbone vector. The *osaA* coding region, along with 1.5 kb of 5′ and 3′UTR were PCR-amplified using the primers osaA_com1 and osaA_com2. The forward primer contained an engineered HindIII site, whereas the reverse primer contained a NotI site. The amplified fragment was digested with the respective restriction enzymes and ligated to the pBC-hygro vector, previously digested with the same enzymes. The pBC-hygro vector includes the selection marker *Hyg*. The recombinant plasmid was named pAD1.1 and was transformed in the ∆*osaA* strain, resulting in the complementation strain (TAD2.1). The strain was confirmed using a diagnostic PCR with primers osaA_OE1 and osaA_OE2.

### 2.5. Construction of the osaA Overexpression Strain

A strain overexpressing *osaA* (OE) was conducted as follows: the coding region of *osaA* was PCR-amplified from *A. fumigatus* genomic DNA using the primers osaA_OE1 and osaA_OE2 with AscI and NotI engineered restriction sites, respectively. The 1443 bp PCR product was then digested using the corresponding enzymes and ligated into the pSSP1.1 plasmid, previously digested with the same enzymes. The pSSP1.1 vector includes the *A. nidulans gpdA* promoter and *trpC* terminator, along with *A. fumigatus pyrG*. The resulting recombinant plasmid (pAD3.1) was transformed into the CEA17 strain. Transformants were confirmed using a diagnostic PCR with the primers osaA_OE1 and A fum pyrG_R.

### 2.6. Morphological Analysis

To examine the role of *osaA* in colony growth, spores of wild-type, Δ*osaA*, Com, and OE strains were point-inoculated on GMM and incubated at 37 °C in the dark. Colony growth was estimated as the colony diameter, which was measured every two days. The experiment was carried out with four replicates.

For an assessment of the effect of *osaA* on conidiation, the same *A. fumigatus* strains were top-agar-inoculated on GMM (5 × 10^6^ spores/plate) and incubated at 37 °C in the dark. Cores were harvested from each plate at 24 h, 48 h, and 72 h after inoculation and homogenized in water. Conidia were quantified using a haemocytometer (Hausser Scientific, Horsham, PA, USA) and a Nikon Eclipse E-400 bright field microscope (Nikon, Inc., Melville, NY, USA). The experiment was performed in triplicates. Also, the spore size was measured under the Nikon E-400 microscope. Measurements of spore diameter (n = 40) were taken.

To evaluate whether *osaA* affects the spore germination rate in *A. fumigatus*, approximately 10^6^ spores/mL of each of the strains were grown in liquid-shaken cultures containing 50 mL of GMM at 250 rpm. After 4 h post-inoculation, the cultures were observed under a bright field microscope, using a haemocytometer, every 2 h. This experiment was carried out in triplicates.

### 2.7. Environmental Stress Tests

To test sensitivity to a range of temperatures, wild-type, Δ*osaA*, Com and OE strains were point-inoculated on GMM and incubated in the dark at 25 °C, 30 °C, 37 °C, and 42 °C for 5 days when photographs and colony diameter measurements were taken. This experiment included three replicates.

To evaluate the importance of *osaA* in the resistance to oxidative stress, the strains were point-inoculated on solid GMM supplemented with 0, 15, 20, 25, or 30 μM menadione and incubated at 37 °C. Photographs were taken after an incubation period of 5 days. The experiment was performed with three replicates.

### 2.8. Cell Wall Tests

To elucidate possible alterations of cell wall integrity due to changes in the *osaA* locus, the wild-type, and Δ*osaA*, Com, and OE strains were point-inoculated in a 24-well plate on solid GMM supplemented with different concentrations (0, 0.005, 0.01, 0.015, and 0.02%) of sodium dodecyl sulfate (SDS) [65], and incubated at 37 °C for 72 h. In a separate experiment, the strains were inoculated on GMM with Congo red at concentrations of 0, 5, 10, 15, 50, or 100 μg/mL and incubated at 37 °C. Photographs were also taken at 72 h after inoculation. The experiments were carried out with three replicates.

Whether osmotic stabilization allowed the recovery of the wild-type phenotype in strains with an altered *osaA* locus and possible cell wall defects was assessed as follows. The conidia (10^6^ spores/mL) of each strain were point-inoculated on GMM supplemented with either 0.6 M of KCl, 1 M of sucrose, 0.7 M of NaCl, or 1.2 M of sorbitol, and incubated at 37 °C for 72 h when photographs were taken, and colony diameters were measured. This experiment was performed in triplicate.

### 2.9. Adhesion Assay

To assess the role of *osaA* in adhesion capacity, each strain was inoculated in 50 mL of GMM (10^6^ spores/mL). Aliquots (130 μL) from these suspensions were then added to each well of a 96-well microtiter plate. Cultures were incubated at 37 °C for a period of 24 h and 48 h. The mycelial mat and supernatant were removed from the wells. Biomass adhered to the well walls and was washed three times with water and then stained with 130 μL of 0.01% Crystal Violet (in water) for 20 min at room temperature. The stained wells were washed again three times with water, allowed to dry, and then de-stained using 130 μL of 30% acetic acid. The absorbance was measured at 560 nm on an Epoch spectrophotometer (Biotek, Winooski, VT, USA).

### 2.10. Metabolomics Analysis

For these analyses, the wild-type, Δ*osaA*, Com, and OE strains were top-agar-inoculated on GMM (5 × 10^6^ spores/plate) and cultured at 37 °C for 72 h. Three cores (16 mm diameter) were harvested from each plate and extracted with chloroform. The strain set was also grown in a liquid yeast extract–glucose–supplements (YES) medium [66] and cultured at 37 °C and 250 rpm for 7 days. The filtrate (30 mL) was collected and extracted with chloroform. Extracts were analyzed using liquid chromatography coupled to high-resolution mass spectrometry (LC-HRMS). Samples were dissolved in 500 μL of an acetonitrile/water mixture (50:50, *v*/*v*) and shaken vigorously for 30 s, and then treated with a sonicator (Bransonic 221 Ultrasonic bath, Emerson Electric, St. Louis, MO, USA) for 2 h. A volume of 250 μL of pure acetonitrile was added to each sample, and vigorously shaken for 30 s and centrifuged (pulse). LC-MS analyses were performed using Acquity HPLC (Waters, Saint-Quentin-en-Yvelines, France) linked to an LTQ Orbitrap XL high-resolution mass spectrometer (Thermo Fisher Scientific, Les Ulis, France). A reversed-phase Luna^®^ C18 column (150 mm × 2.0 mm, 5 μm) (Phenomenex, Torrance, CA, USA) was utilized with water acidified with 0.1% formic acid and 100% acetonitrile as the mobile phases. The elution gradient was as follows: 0–30 min, B: 20–50%; 30–35 min, B: 50–90%; 35–45 min, B: 90%; 45–50 min, B: 90–20%; and 50–60 min, B: 20%. The flow rate was 0.2 mL min^−1^, the column temperature was 30 °C, and a volume of 10 μL was injected. HRMS acquisitions were carried out with electrospray ionization (ESI) in positive and negative modes, as previously reported [67]. MS/MS spectra were obtained with the CID mode at low resolution and a collision energy of 35%.

### 2.11. Gene Expression Analysis

Total RNA was obtained from lyophilized mycelial samples using TRIsure^TM^ (Meridian Bioscience, Bioline, Cincinnati, OH, USA), following the manufacturer’s instructions. Gene expression was assessed using qRT-PCR. Ten micrograms of total RNA was treated with the Ambion Turbo DNA-free Kit (ThermoFisher Scientific, Waltham, MA, USA). cDNA was synthesized with Moloney murine leukemia virus (MMLV) reverse transcriptase (Promega, Madison, WI, USA). qRT-PCR was performed with the BioRad CFX96 Real-Time PCR System or Applied Biosystems 7000, using Q SYBR green Supermix (BioRad, Hercules, CA, USA) or SYBR green Jumpstart *Taq* Ready mix (Sigma, St. Louis, MO, USA) for fluorescence detection. The primer pairs used for qRT-PCR are listed in Appendix A. The gene expression data for each gene were normalized to that of the *A. fumigatus* histone H4 unless specified otherwise. Relative expression levels were calculated using the 2^−ΔΔCT^ method [68].

### 2.12. Virulence Assay

The inoculum was generated as follows: all strains were inoculated on stabilized minimal medium (GMM + 1.2 M sorbitol) and grown at 37 °C for 5 days. Conidia were collected by flooding the plates with 0.01% Tween-80 and scraping them with sterile cotton tip applicators. The conidia were filtered through Miracloth, washed twice with sterile water, and resuspended in sterile phosphate-buffered saline (PBS, ThermoFisher Scientific, Waltham, MA, USA).

Outbred female CD-1 mice (Charles River), six to eight weeks old, weighing 18–24 g, were used for the virulence experiments. For a neutropenic mouse model, immunosuppression was carried out via intraperitoneal injection with 175 mg/kg cyclophosphamide (Baxter, Deerfield, IL, USA) two days before inoculation [69], followed by a subcutaneous injection 40 mg/kg of triamcinolone acetonide (Kenalog-10 from Bristol-Myers Squibb, Lawrenceville, NJ, USA) one day before inoculation with *A. fumigatus* spores. Immunosuppressed mice were briefly anesthetized using a small rodent anesthesia machine and inoculated with *A. fumigatus* wild-type, Δ*osaA*, *osaA* Com, or *osaA* OE in the concentration of 2 × 10^6^ spores in 40 µL of PBS. A control group of five mice were immunosuppressed and injected with PBS. All the mice were monitored three times daily from days 1 to 7 and once daily from days 8 to 14. Injections with cyclophosphamide and Kenalog were repeated on days +3 and +6, respectively. A statistical comparison of the associated Kaplan–Meier curves was conducted with the log-rank test using GraphPad Prism (version 8.0).

For the corticosteroid-induced immune suppression mouse model, mice were immunosuppressed with a single 40 mg/kg injection of Kenalog-10 one day before infection with *A. fumigatus* conidia. Mice were monitored three times daily from days 1 to 7 and once daily from days 8–14. A statistical comparison of the associated Kaplan–Meier curves was also conducted with the log-rank test using GraphPad Prism (version 8.0).

This study was carried out in strict accordance with the Guide for the Care and Use of Laboratory Animals of the National Research Council. The protocol was approved by the Institutional Animal Care and Use Committee of Clemson University (Permit #AUP2022-0236). All efforts were made to minimize suffering. Humane euthanasia via CO_2_ inhalation was performed when mice met the criteria indicating a moribund state; these endpoints include behaviors of unresponsiveness to tactile stimuli, inactivity, lethargy, staggering, anorexia, and/or clinical signs of bleeding from the nose or mouth, labored breathing, agonal respirations, a purulent exudate from the eyes or nose, abnormally ruffled fur, or greater than 20% weight loss. The method of euthanasia via CO_2_ inhalation is consistent with recommendations of the Panel on Euthanasia of the American Veterinary Medical Association.

### 2.13. Statistical Analysis

Statistical analysis was carried out using analysis of variance (ANOVA) in conjunction with Tukey’s multiple test comparison test, unless stated otherwise.

## 3. Results

### 3.1. Identification of OsaA in A. fumigatus

The OsaA protein (corresponding to AFUB_093810) of *A. fumigatus* is conserved with respect to homologs in other species of the genus *Aspergillus* (Figure 1 and Figure 2, and Table 2). Our BLASTP analysis demonstrated that *A. fumigatus* OsaA presents the highest similarity to *Aspergillus fischeri* (98.96% percent identity, accession number: XP_001257654.1), followed by the homolog in *Aspergillus flavus* (82.32% identity, accession number: AFLA_004600). The OsaA in *A. nidulans* (AN6578) showed a 64.24% identity with OsaA in *A. fumigatus* (Table 2). Our BLASTP analysis also revealed the high conservation of OsaA with other species within Ascomycota, beyond *Aspergillus*. The sequence alignment of *A. fumigatus* OsaA with other fungal homologs showed the conservation of the WOPR domain (Figure 1). The WOPR domain in *A. fumigatus* OsaA was divided into two regions (9–94 and 158–222 residues), connected to each other by an interlinked sequence. Conserved domain analysis revealed the presence of a Gti1/Pac2 domain (Accession pfam09729) in the N-terminal region of the protein. Members of this family are orthologs of *Saccharomyces pombe*, such as the Pac2 WOPR fragment [56].

### 3.2. Growth, Conidiation, and Germination Rate Are Regulated by osaA in A. fumigatus

To determine the function of *osaA* in *A. fumigatus*, an *osaA* deletion, Δ*osaA*, as well as complementation and overexpression strains were generated (Appendix A). A diagnostic PCR confirmed the gene replacement of *osaA* with the *pyrG* marker and obtained the expected 2.64 kb product. PCR was also used to verify the integration of the wild-type allele in the Δ*osaA* strain in the Com strain, amplifying a 1.4 kb product (Appendix A). In addition, *osaA* overexpression strains were confirmed via a PCR amplifying the expected 1.97 kb product.

An evaluation of the colony growth revealed a significant reduction in colony diameter in the *osaA* deletion mutant compared to the wild-type (Figure 3). The back of the mutant colonies showed dark pigmentation, which was not present in the wild-type colonies. The complementation of Δ*osaA* with the *osaA* wild-type allele recovered the wild-type phenotype. The constitutive overexpression of *osaA* also reduced colony growth but to a lesser extent. The back of the *osaA* overexpression strain colony exhibited a yellowish pigmentation and distinct furrowing when compared to the wild-type colony.

Conidia are the inoculum for *A. fumigatus* infections. Previous studies in *A. nidulans* showed that the production of these asexual air-borne spores was affected by *osaA* in the model fungus [61]. In the present study, the role of *osaA* in conidiation was assessed in *A. fumigatus*. The Δ*osaA* and OE strains showed a significant reduction in conidial production compared to the wild-type and Com strain, particularly in the deletion strain (Figure 4). Our experiments revealed that both the deletion and overexpression of *osaA* resulted in alterations in expression levels of the transcription factor genes *brlA*, *abaA*, and *wetA* of the central regulatory pathway leading to conidiation [70], particularly in the absence of *osaA*, resulting in a significant reduction (Figure 4). Interestingly, micrographs of Δ*osaA* spores indicated a significant enlargement of conidial size (57%) compared to spores produced by the wild-type. Furthermore, a reduction in germination rate was observed in the absence of *osaA*, while the overexpression of *osaA* resulted in precocious germination (Figure 4).

### 3.3. osaA Influences A. fumigatus Temperature and Oxidative Stress Resistance

Our results revealed that the absence of *osaA* alters *A. fumigatus* thermotolerance (Figure 5). The *osaA* gene is required for growth at 42 °C. This is relevant since the pathogenic nature of *A. fumigatus* is attributed to its thermotolerance, allowing this fungus to survive in a warmer environment and to combat human inflammatory responses. The colony growth defect of this mutant is slightly remediated at temperatures lower than 37 °C. We also noted that the knockout mutant showed increased pigmentation compared to the wild-type, particularly when grown at 25 °C and 30 °C.

An important component in the host’s defense against *A. fumigatus* infections includes the formation of an oxidative stress environment [16,71]. In our study, either the absence or constitutive overexpression of *osaA* both resulted in increased sensitivity to oxidative stress. Although at 10 μM of menadione, the deletion strain colony grew slightly healthier, both deletion and overexpression *osaA* strain colonies were unable to grow in the presence of 30 μM of menadione, a condition that still allowed for growth in wild-type and Com colonies (Figure 6).

### 3.4. The Effect of osaA on A. fumigatus Cell Wall and Adhesion Capacity

The fungal cell wall is the first point of contact with the host. To elucidate whether *osaA* influences cell wall integrity, we exposed the deletion and overexpression of *osaA* strains, as well as their controls, to cell wall stressors. When grown on a medium supplemented with SDS, the growth of the *osaA* deletion mutant was compromised (Appendix A). In addition, when Congo red was supplemented to the medium, the growth of Δ*osaA* was severely affected at a concentration of 5 μg/mL and was unable to grow at 10 μg/mL (Figure 7). The growth of the OE strain was also dramatically affected at 25 μg/mL of Congo red.

The presence of osmotic stabilizers (0.6 M KCl, 1 M sucrose, 0.7 M NaCl, or 1.2 M sorbitol) in the medium allowed for the healthier growth of the deletion *osaA* mutant than in their absence, although the colony growth was still significantly smaller than that of the wild-type growing under the same experimental conditions (Appendix A).

Biofilm formation in the host is relevant in *A. fumigatus* resistance to anti-fungal drugs and to evade the host immune system. Importantly, adhesion to surfaces is required for the formation of biofilm [60]. Our analysis indicated a delay in the capacity to bind to abiotic surfaces in the absence of *osaA* at 24 h after inoculation (Appendix A). At 48 h, the adhesion capacity in Δ*osaA* increased over time; however, it was still significantly lower with respect to that of the wild-type. Interestingly, the overexpression of *osaA* resulted in a significant loss of adhesion capacity after 48 h.

### 3.5. osaA Regulates Secondary Metabolism in A. fumigatus

*Aspergillus fumigatus* produces numerous bioactive compounds that serve multiple functions, including those involved in pathogenicity. Our metabolomics analysis revealed that *osaA* positively regulated secondary metabolism, including the production of fumiquinazoline C, helvolic acid, pyripyropene A, and fumagillin when growing on GMM (Figure 8). In addition, the deletion of *osaA* completely abolished the production of pseurotin A (Appendix A) when growing on the same medium. In the case of the OE strain, the production of pseurotin A was significantly overproduced compared to the wild-type under these culture conditions (Appendix A). When the YES medium was used (Table 3), the *osaA* deletion mutant presented a reduction or was unable to produce the above-mentioned metabolites, along with helvolinic acid, 1,2 dihydrohelvolic acid and gliotoxin, which were absent in the deletion mutant. The overexpression of *osaA* in YES medium leads to a significant decrease or lack of production of fumiquinazoline C, helvolinic acid, 1,2-dihydrohelvolic acid, fumagillin, and gliotoxin detected in the wild-type strain under the same experimental conditions.

### 3.6. Gene Expression Analysis of Secondary Metabolite Genes

Genes from gene clusters associated with the secondary metabolites whose synthesis was found to be *osaA*-dependent were selected as indicators of cluster activation. Their expression was analyzed in the wild-type, Δ*osaA*, Com, and OE strains using qRT-PCR. Our results indicated that *osaA* is necessary for the normal transcription of the *fmqD* gene, encoding a FAD-dependent oxidoreductase in the fumiquinazoline C biosynthetic gene cluster [72]. The transcription of the *pdsA* cyclase gene, involved in the synthesis of helvolic acid [73], and *pypC*, necessary for the synthesis of pyripyropene A [74], were significantly downregulated in the deletion strain. In addition, the transcription factor gene, *fumR* [75], involved in the regulation of other genes in the fumagillin gene cluster, displayed a considerable loss of expression when *osaA* was deleted (Figure 8). Furthermore, our study revealed that *osaA* was required for the normal transcription of *psoA*, encoding a polyketide synthase, in the pseurotin A gene cluster [65,76] (Appendix A).

Since gliotoxin production was reduced in the Δ*osaA* strain, we also examined whether the expression of key genes in the gliotoxin gene cluster was *osaA*-dependent. Our results indicated a drastic reduction in the expression of the Zn(II)2Cys6 binuclear transcription factor *gliZ* [77] in the absence of *osaA* with respect to the wild-type control (Appendix A). In addition, the expression of *gliP*, encoding a nonribosomal peptide synthase [78], was also downregulated in the *osaA* deletion strain. Interestingly, our results also showed that the expression of the global regulatory genes *veA* and *laeA* is *osaA*-dependent (Figure 9).

### 3.7. osaA Is Indispensable for Normal A. fumigatus Virulence in Both Neutropenic and Corticosteroid-Immunodepressed Murine Models

Since *A. fumigatus osaA* affects multiple cellular processes in this fungus, including those influencing pathogenicity, we tested whether *osaA* is involved in virulence (Figure 10). Importantly, the groups infected with spores from the deletion *osaA* mutant showed higher survival rates when compared to the groups infected with wild-type spores, indicating that *osaA* is indispensable for virulence in either neutropenic or corticosteroid immune-repressed environments. The complementation strain presented a similar virulence pattern to that of the wild-type strain in both experiments, indicating the recovery of the wild-type phenotype when the *osaA* wild-type allele was reintroduced in the deletion mutant. The overexpression of *osaA* resulted in a slight reduction in the mortality rate in the neutropenic model, while, in the corticosteroid model, OE presented a similar mortality rate to that of the wild-type.

## 4. Discussion

Transcriptional regulators with a WOPR domain are found across different fungal genera [79] and have been known to act as master regulators governing morphological development, sporogenesis, and pathogenesis [55,56,57,58,59,60]. Our work shows that OsaA in *A. fumigatus* includes a WOPR domain in the N-terminal region. This domain is present in two sub-regions and interconnected with a short linker region, as is the case in the same conserved domain in the *C. albicans* Wor1 protein [55,79,80].

In closely related dimorphic fungi, *H. capsulatum* and *Coccidioides* spp., the WOPR protein Ryp1 is involved in regulating host responses to temperature that result in a switch between two phenotypic forms necessary for the organism’s commensalism [81,82]. Specifically, in the absence of Ryp1, the mycelial form of *H. capsulatum* is genetically dominant at 37 °C [57], whereas, in *Coccidioides*, it maintains genetic development in the environmental hyphae form as well as the pathogenic spherulated form [81]. In *A. nidulans*, the role of *osaA* was identified to be a crucial factor in development. Our analysis also indicated that although *A. fumigatus osaA* presents commonalities regarding its function in other species, including *A. nidulans*, it also presents some distinct roles. For example, as in *A. nidulans* [61], *osaA* is important for proper conidiation in *A. fumigatus*; however, *osaA* also has a role in germination and vegetative growth in this opportunistic pathogen.

Importantly, in this study, we demonstrated that *osaA* is indispensable for virulence in both neutropenic and corticosteroid-induced murine models. The pathogenicity of *A. fumigatus* is associated with multiple virulent factors; some of these factors are affected by *osaA*. Our microscopic observation indicated that the spore size of the deletion *osaA* strain is larger than that of the wild-type strain. The small spore size of *A. fumigatus* allows them to easily enter the respiratory tract of humans and other animals. Spore enlargement in the absence of *osaA* could reduce the capability to reach the lung alveoli, reducing systemic infection [37,83,84]. In addition, decreased thermotolerance in a strain with an *osaA*-altered locus could negatively impact its ability to grow inside mammalian lungs [85,86]. Recent studies have also indicated that thermotolerance regulatory genes play a vital role in the *A. fumigatus* cell wall integrity pathway [86]. Our results revealed that both the deletion and overexpression of *osaA* (particularly the former) are more susceptible to cell wall stressors, suggesting a loss of integrity. The addition of osmotic stabilizers to the deletion of *osaA* cultures partially restored colony growth, further supporting the role of *osaA* in the cell-wall function. Importantly, *osaA* affects the capacity of the fungus to adhere to surfaces, a condition necessary for the formation of biofilm, which is crucial for fungal pathogenicity, protecting the fungus from external stresses, including antifungal agents [40,87,88].

During *A. fumigatus* infection, the host immune system elicits an elevated production of the macrophage and neutrophil-mediated killing of the conidia, one of which is through the generation of reactive oxygen species (ROS) [16]. Here, we also explored whether *osaA* is necessary for resistance to oxidative stress in *A. fumigatus*. Our results showed an increase in sensitivity to high levels of menadione in the deletion and overexpression of *osaA* strains compared to the wild-type. The addition of menadione in the medium simulates the production of ROS inside the cell [89,90], thus creating an oxidative stressful condition. This higher sensitivity to ROS in *A. fumigatus* could influence the survival of the infected host.

In addition, *A. fumigatus* is known to produce numerous secondary metabolites that impact host defenses at different stages [91,92]. These compounds are known to either suppress or avoid immune sentinel cells [93]. Our metabolomics analysis revealed a profound effect of *osaA* in *A. fumigatus* secondary metabolism. For example, pseurotin A and fumagillin, both associated with the same biosynthetic gene cluster [57,94], are not produced in the *osaA* mutant. This is relevant, as fumagillin alters neutrophil activity [95], and pseurotin A is an antioxidant [96]. It is possible that the lack of pseurotin A in the *osaA* mutant could contribute to the observed higher sensitivity to ROS in this strain. Other compounds not produced in the *osaA* deletion strain and produced at significantly lower levels when *osaA* is overexpressed are helvolic acid, a cilioinhibitor, and the derivatives helvolinic acid and 1,2-dihydrohelvolic acid, which presents antibacterial properties against *Streptococcus aureus* [97], fumiquinazoline, which presents cytotoxic activity, and pyripyropenes, an acyl-CoA/cholesterol acyltransferase inhibitor [98,99,100,101]. Furthermore, the production of gliotoxin was severely compromised when *osaA* was differentially expressed in *A. fumigatus*. Gliotoxin’s role in *A. fumigatus* infection has been previously studied in depth [8,16,93,102], where it was shown to affect host macrophage phagocytosis and to inhibit the activity of other immune cells [93].

The *osaA* gene was required for the expression of numerous genes in biosynthetic gene clusters associated with the production of secondary metabolites; for instance, *gliZ* and *gliP* [77,78], involved in the production of gliotoxin, and *fmqD*, *pdsA*, *pypC*, and *fumR*, involved in the synthesis of fumiquinazoline C, helvolic acid, pyripyropene, and fumagillin, respectively [65,72,73,75,76,103] were downregulated in the *osaA* deletion mutant with respect to the wild-type. Based on our results, the effect of *osaA* on the expression of these gene clusters could be, at least in part, mediated by *osaA*’s influence on the expression of the global regulatory genes *veA* and *laeA*. VeA and LaeA are components of the *velvet* protein complex [16,48], which governs several biological processes in numerous fungi, including secondary metabolism and development [48,74,104,105,106,107,108,109]. In *A. nidulans*, *osaA* has been shown to regulate the developmental balance between conidiation and sexual reproduction in a *veA*-dependent manner [61]. In *A. fumigatus*, we observed the downregulation of the expression of the developmental genes *brlA*, *abaA*, and *wetA* in the absence of *osaA*, which was associated with a decrease in conidiation. Based on our expression analysis results, it is likely that the effect of *osaA* on *veA* and *laeA* in *A. fumigatus* could result in multiple outputs, affecting secondary metabolism, development, and sensitivity to oxidative stress, among other biological processes [105].

## 5. Conclusions

In conclusion, *A. fumigatus osaA* is indispensable for virulence in neutropenic and corticosteroid-induced immune-repressed environments. The *osaA* gene influences growth, development, and sensitivity to environmental stressors, such as oxidative and temperature stresses, which could affect pathogenicity in this fungus. Furthermore, *osaA* was found to positively regulate the capacity of adhesion to surfaces, which is an important characteristic for the formation of biofilm. Importantly, this WOPR domain transcription factor gene is also necessary for the normal expression of secondary metabolite gene clusters and the concomitant synthesis of natural products, many of which are known to play a role during infection, including gliotoxin. In addition, *osaA* regulates the expression of *veA* and *laeA* genes, known global regulators governing development and secondary metabolism in *A. fumigatus* and other fungi. This study suggests that *osaA* or its product are promising targets with the potential for use in a control strategy against *A. fumigatus* infections.

## Figures and Tables

**Figure 1 jof-10-00103-f001:**
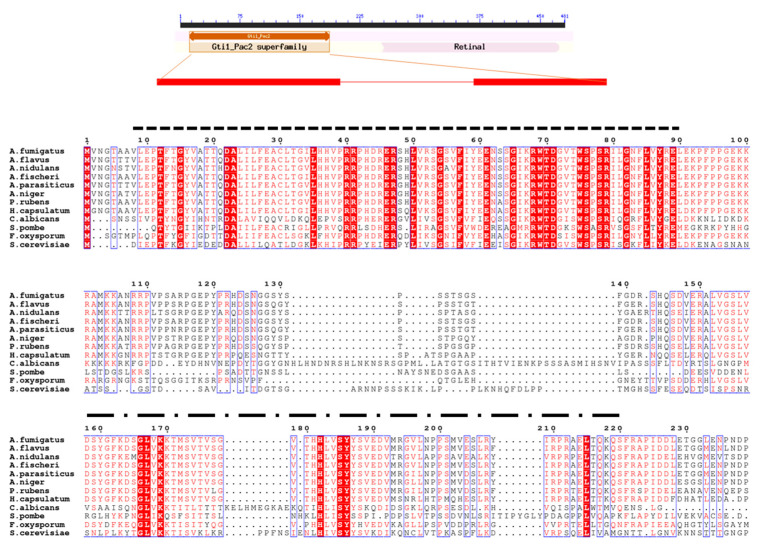
Alignment of OsaA homologs. Alignment of OsaA of *A. fumigatus* (AFUB_093810) with homologs of *A. flavus* (AFLA_004600), *A. nidulans* (AN6578.2), *A. fischeri* (XP_001257654.1), *A. parasiticus* (KAB8200295.1), *A. niger* (XM_001396613.1), *P. rubens* (XM_002563660.1), *H. capsulatum* Ryp1 (ABX74945.1), *C. albicans* Wor1(Q5AP80), *S. pombe* Pac2 (BAC54908.1), *S. cerevisiae* Mit1 (NC_001137), and *F. oxysporum* Sge1 (XM_018389881). Sequences were obtained from NCBI and aligned using Clustal W Multiple Sequence Alignment tool and ESPript. Two conserved regions of WOPR domain were identified as WOPRa (dash-lined) and WOPRb (dash-dot-lined). Red box with white characters indicates strict identical residues, whereas red characters indicate similarity in a group. Blue frame depicts similarity across groups.

**Figure 2 jof-10-00103-f002:**
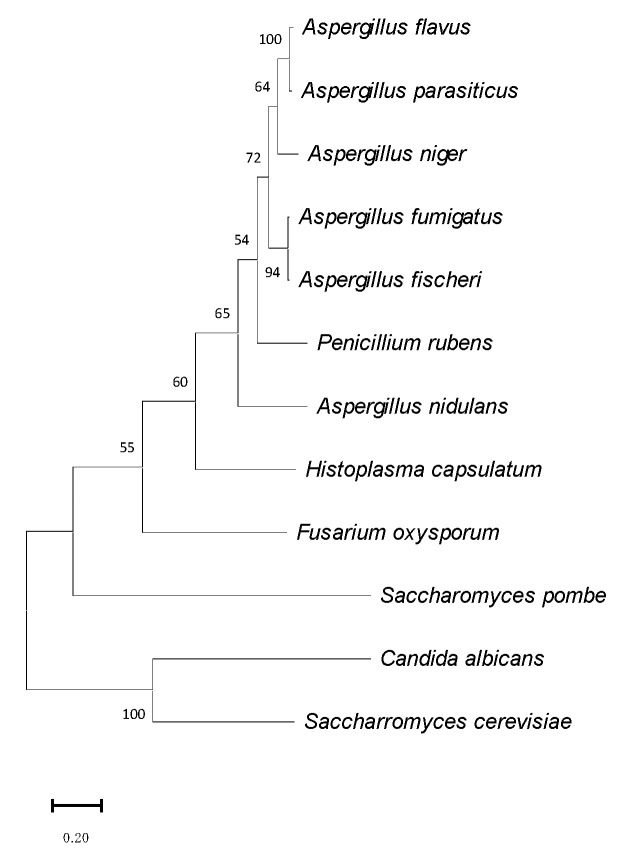
Phylogenetic tree of OsaA homologs from different fungal species. Construction of a phylogenetic tree was carried out using MEGA v11.0. The tree was generated with the maximum likelihood model with a bootstrap value of 1000 (http://megasoftware.net/, 11 June 2023).

**Figure 3 jof-10-00103-f003:**
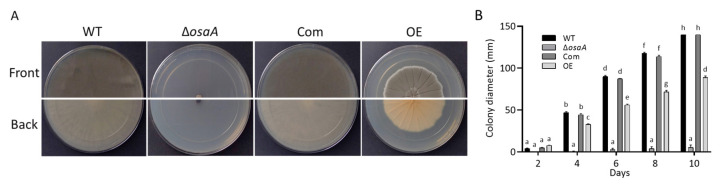
*osaA* influences growth and development in *A. fumigatus*. (**A**) The wild-type (WT), Δ*osaA*, complementation strain (Com), and overexpression strain (OE) were point-inoculated on GMM and grown in the dark at 37 °C. Photographs were taken after 10 days. (**B**) Colony diameter was measured every 2 days. The error bars represent standard errors. Different letters on the columns indicate values that are statistically different (*p* < 0.05), as determined using one-way ANOVA with Tukey’s test comparison.

**Figure 4 jof-10-00103-f004:**
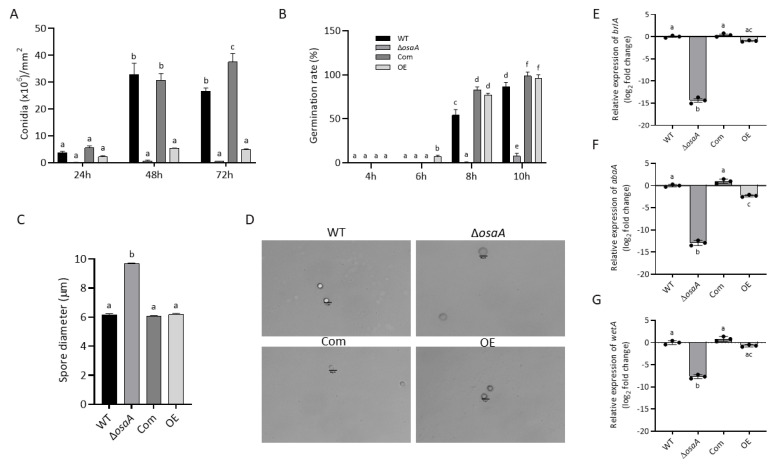
*osaA* regulates conidial production and spore size. (**A**) Conidial quantification of top-agar-inoculated plates grown at 37 °C for 24 h, 48 h, and 72 h. Conidia were observed under a bright field microscope and counted with a hemocytometer. (**B**) Germination rate assessment. Liquid GMM cultures were grown at 37 °C under shaking conditions. Germination was evaluated every 2 h under the microscope using a hemocytometer. (**C**) Measurements of spore diameter (n = 40 for each strain). Error bars represent the standard error. Different letters on the columns indicate values that are statistically different (*p* < 0.05). (**D**) Micrographs of *A. fumigatus* wild-type (WT), Δ*osaA*, *osaA* complementation (Com), and *osaA* overexpression (OE) conidia. The scale bar represents 10 µm. (**E**–**G**) Gene expression analysis of developmental genes *brlA* (**E**), *abaA* (**F**), and *wetA* (**G**). WT, Δ*osaA*, Com, and OE strains were grown in GMM liquid stationary cultures at 37 °C for 48 h. Expression was calculated using the 2^−ΔΔCT^ method [68]. Error bars represent the standard error, and different letters on the bar represent significantly different values (*p* < 0.05). Color key in (**B**) also applies to (**A**–**C**) and (**E**–**G**).

**Figure 5 jof-10-00103-f005:**
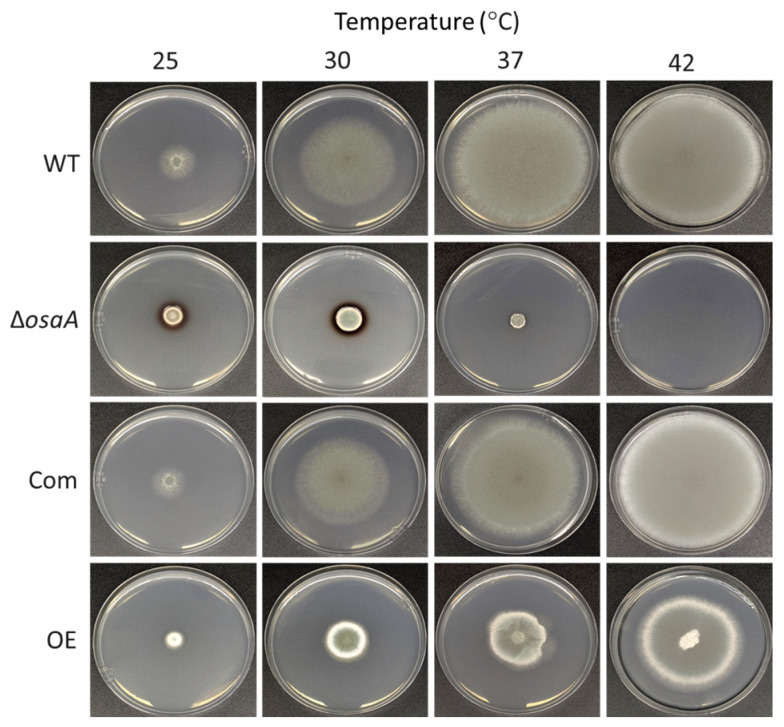
Thermotolerance is affected by *osaA* in *A. fumigatus*. *A. fumigatus* wild-type (WT), Δ*osaA*, complementation (Com), and overexpression (OE) strains were point-inoculated on GMM and incubated in the dark at 25 °C, 30 °C, 37 °C, or 42 °C. Photographs were taken 5 days after inoculation. The experiment was carried out with three replicates.

**Figure 6 jof-10-00103-f006:**
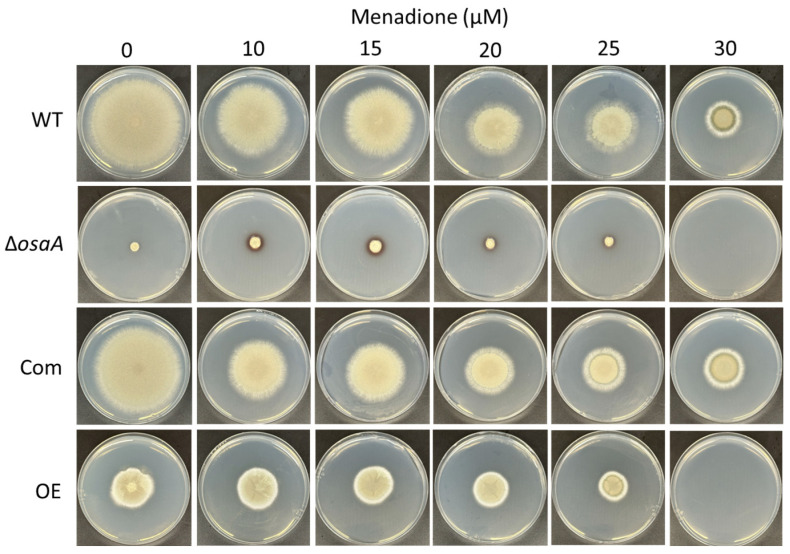
*osaA* is necessary for resistance to oxidative stress. Plates of solid GMM and GMM supplemented with increasing concentrations of menadione (from 0 to 30 µM) were point-inoculated with *A. fumigatus* wild-type (WT), Δ*osaA*, complementation (Com) and overexpression (OE) strains and incubated at 37 °C for 5 days. The experiment was conducted with three replicates.

**Figure 7 jof-10-00103-f007:**
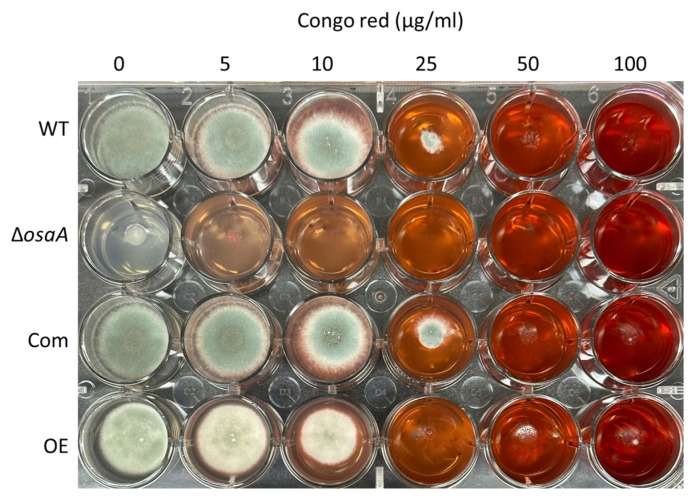
Effect of *osaA* in response to cell wall stress. *A. fumigatus* wild-type (WT), Δ*osaA*, complementation (Com), and overexpression (OE) strains were point-inoculated on GMM and GMM supplemented with different concentrations of Congo red. Plates were incubated at 37 °C for 72 h. The experiment was carried out in triplicates.

**Figure 8 jof-10-00103-f008:**
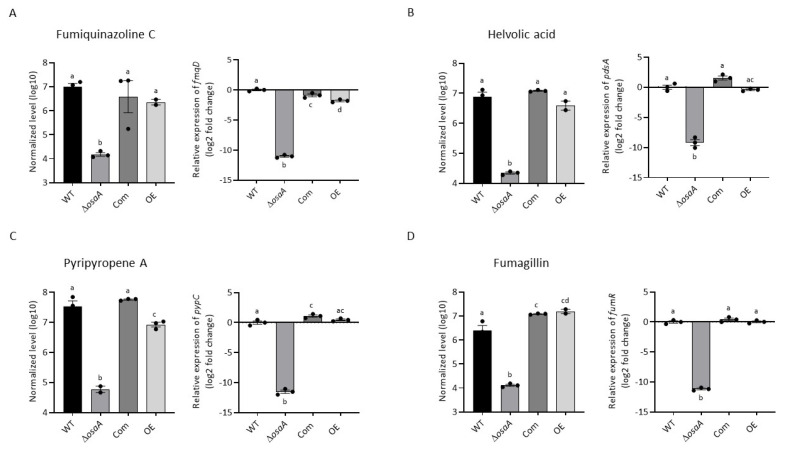
*osaA* regulates the production of (**A**) fumiquinazoline C, (**B**) helvolic acid, (**C**) pyripyropene A and (**D**) fumagillin in *A. fumigatus.* The *A. fumigatus* wild-type (WT), Δ*osaA*, complementation (Com), and overexpression (OE) strains were top-agar-inoculated on GMM and incubated at 37 °C for 72 h. Extracts were analyzed using liquid chromatography coupled to high-resolution mass spectrometry (LC-HRMS). On the right in each panel, the expression of key genes in the corresponding secondary metabolite genes clusters, *fmqD*, *pdsA*, *pypC*, and *fumR*, respectively, were selected as indicators of cluster activation. Samples were obtained from GMM liquid stationary cultures incubated for 48 h. Expression was calculated following the 2^−ΔΔCT^ method [68]. Error bars represent the standard error. Different letters on the columns indicate values that are statistically different (*p* < 0.05), as determined using one-way ANOVA with Tukey’s test comparison. All the experiments were performed in triplicates.

**Figure 9 jof-10-00103-f009:**
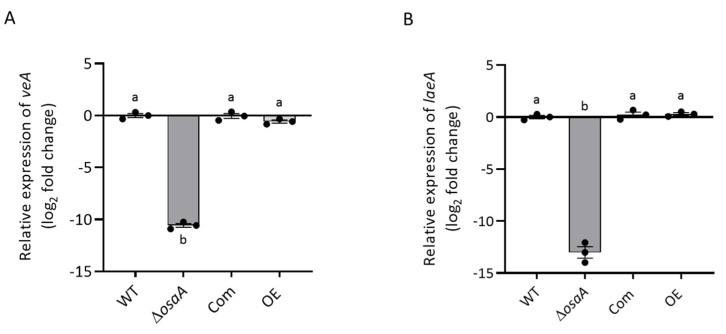
*osaA* affects the gene expression of the global regulators *veA* and *laeA* in *A. fumigatus*. *A. fumigatus* wild-type (WT), Δ*osaA*, complementation (Com) and overexpression (OE) strains were grown on GMM stationary cultures for 48 h. The relative gene expression of (**A**) *veA* and (**B**) *laeA* was calculated using the 2^−ΔΔCT^ method [68]. Error bars represent the standard error. Different letters on the columns indicate values that are statistically different (*p* < 0.05).

**Figure 10 jof-10-00103-f010:**
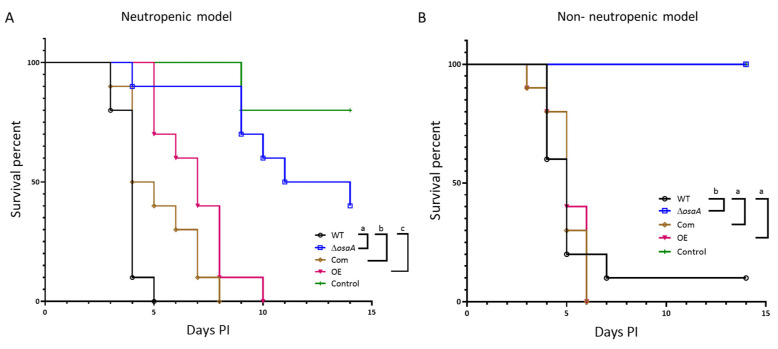
*osaA* is indispensable for *A. fumigatus* virulence in both neutropenic and corticosteroid-induced immune-repressed murine models. The *A. fumigatus* wild-type (WT), Δ*osaA*, complementation (Com), and overexpression (OE) strains (2 × 10^6^ spores/40 μL PBS) were used to infect female outbred CD-1 mice. (**A**) For the neutropenic model, mice received an intraperitoneal injection of cyclophosphamide and Kenalog, as detailed in the Section 2. Uninfected controls were treated with cyclophosphamide and Kenalog but not fungal spores. Post-infection, mice were observed three times daily for the first 7 days and once daily from days 8 to 14. Each group included 10 mice. Statistical analysis was performed utilizing GraphPad PRISM and the Mantel–Cox test. Different letters indicate that values are statistically different. Survival was significantly different between Δ*osaA* and WT (*p* < 0.0001) and OE and WT (*p* < 0.0001). (**B**) For the non-neutropenic model, mice received only Kenalog one day prior to the infection. Uninfected controls were treated with Kenalog but not fungal spores. Mice were monitored for survival for 14 days. Statistical analysis was performed using GraphPad PRISM statistics. This was significantly different between Δ*osaA* and WT (*p* < 0.0001) and between OE and WT (*p* ≤ 0.001).

**Table 1 jof-10-00103-t001:** Strains used in this study.

Strain Name	Genotype	Source
CEA10	Wild-type	Gift from Robert Cramer
CEA17	*pyrG1*	Gift from Robert Cramer
TAD1.1	*pyrG1*Δ*osaA*::*pyrGAparasiticus*	This study
TAD2.1	*pyrG1* Δ*osaA*::*pyrG osaA*::*hyg*	This study
TSSP41.1	*gpdA*::*osaA*::*trpC*::*pyrG*	This study

**Table 2 jof-10-00103-t002:** Percentage of identity and similarity of *A. fumigatus* OsaA with homologous proteins in species within the Ascomycota phylum.

Accession Number	Organism Name	% Identity	% Similarity
RAQ53329.1	*Aspergillus flavus*	82.32	89.5
AN6578.2	*Aspergillus nidulans*	64.24	73.1
XP_001257654.1	*Aspergillus fischeri*	98.96	99.4
KAB8200295.1	*Aspergillus parasiticus*	82.32	89.7
XM_001396613.1	*Aspergillus niger*	82.42	88.4
XM_002563660.1	*Penicillium rubens*	72.57	80.4
ABX74945.1	*Histoplasma capsulatum*	58.13	64.9
Q5AP80	*Candida albicans*	30.02	24.1
BAC54908.1	*Saccharomyces pombe*	35.45	25.5
XM_018389881	*Fusarium oxysporum*	44.16	38.9
NC_001137	*Saccharomyces cerevisiae*	30.00	33.5

**Table 3 jof-10-00103-t003:** LC-HRMS analysis data from YES cultures (average of normalized values ± standard error).

Compound	WT	Δ*osaA*	Com	OE
Fumiquinazoline C	2.58 × 10^7^ ± 2.80 × 10^6^	1.43 × 10^4^ ± 1.06 × 10^3^	2.99 × 10^7^ ± 2.99 × 10^6^	2.92 × 10^6^ ± 1.38 × 10^6^
Pyripyropene A	1.20 × 10^8^ ± 1.86 × 10^7^	1.02 × 10^6^ ± 5.52 × 10^5^	1.15 × 10^8^ ± 3.14 × 10^7^	1.88 × 10^7^ ± 1.16 × 10^7^
Helvolic acid	9.55 × 10^6^ ± 2.70 × 10^6^	Not detected	1.50 × 10^7^ ± 1.34 × 10^6^	2.96 × 10^6^ ± 1.50 × 10^6^
Helvolinic acid	3.59 × 10^6^ ± 3.96 × 10^5^	Not detected	5.03 × 10^6^ ± 6.99 × 10^5^	8.91 × 10^5^ ± 4.01 × 10^5^
1,2-dihydrohelvolic acid	1.88 × 10^6^ ± 4.36 × 10^5^	Not detected	3.11 × 10^6^ ± 5.14 × 10^5^	5.34 × 10^5^ ± 2.14 × 10^5^
Fumagillin	2.27 × 10^7^ ± 1.74 × 10^6^	Not detected	2.76 × 10^7^ ± 4.85 × 10^6^	8.59 × 10^6^ ± 2.83 × 10^6^
Gliotoxin	7.50 × 10^4^ ± 1.55 × 10^4^	Not detected	7.01 × 10^4^ ± 1.47 × 10^4^	Not detected

## Data Availability

Data are contained within the article and Appendix A.

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
