# Peer review of "Role of the osaA Gene in Aspergillus fumigatus Development, Secondary Metabolism and Virulence"

_jof, 2024, doi:10.3390/jof10020103_

Round 1

Reviewer 1 Report

Comments and Suggestions for Authors

The manuscript presents a very complete work on the transcription factor OsaA in the pathogen Aspergillus fumigatus. It contains an introduction that reviews the problem of this pathogen to different types of patients as well as the problem of resistant isolates to azole treatment. Moreover, the introduction include background about the previous studies on the orthologue to OsaA in other yeasts and filamentous fungi, that include the pathogens Candida albicans and Fusarium graminearum.

The experiments are well designed, and the results showed very clear differences among the strains in different aspects which indicates that this gene is involved in the physiology of the fungus (conidiation, secondary metabolite production), response to cell wall stress, production of secondary metabolites, and virulence to mammals.

The results support the conclusions. This gene is indispensable for virulence because it influences important aspects for pathogenicity as growth and response to stress.

It is a work of great importance for the regulation of the most important aspects in fungi, and I believe that it will be of great interest to the scientific community.

The figures containing both photographs and graphs are of high quality and statistics has been included in the graphs.

The italics in genes and Latin names are missing in the introduction and results sections.

Author Response

We greatly appreciate the comments of Reviewer 1 and Reviewer 2. The suggested changes have been incorporated in the improved revised version.

Reviewer1:

The italics in genes and Latin names are missing in the introduction and results sections.

***Thank you very much for your comments.  It seems  that  italics were lost during submission.  We have fixed this problem in the following locations in the revised manuscript.  These are also indicated in the “Track changes” copy of this document, indicated in blue font.

Line   68: in vivo

          281: Aspergillus fischeri

          282: Aspergillus flavus

          283: A. nidulans

          285: Aspergillus

          291: Saccharomyces pombe

          311: pyrG 

          329: A. nidulans

          335, 351: brlA, abaA, wetA

          327, 349, 354, 430, 460, 483, 484, 487: P

          407: Aspergillus fumigatus 

          427: fmqD, pdsA, pypC, and fumR

          438: fmqD

          440: pdsA

          441: pypC

          442: fumR

          445: psoA

          450: gliZ

          451: gliP

          453, 456, 458: veA, laeA

          278, 280, 284, 286, 287, 309, 323, 329, 331, 349, 356, 358, 364, 368, 378, 391, 399, 423, 456, 475, 476 : A. fumigatus 

          309, 311, 312, 313, 316, 318, 319, 320, 323, 330, 331, 332, 334, 336, 337, 340, 343, 349, 350, 351, 356, 357, 364, 370, 372, 376, 378, 383, 385, 387, 391, 396, 402, 403, 404, 409, 411, 415, 417, 422, 423, 436, 437, 438, 444, 445, 447, 448, 450, 452, 453, 456, 457, 463, 464, 465, 467, 470, 471, 475, 476, 483, 487 : osaA

          279, 285: Aspergillus

Reviewer 2 Report

Comments and Suggestions for Authors

Title: Review of "Role of the osaA Gene in Aspergillus fumigatus Development, Secondary Metabolism, and Virulence"

The manuscript entitled "Role of the osaA gene in Aspergillus fumigatus development, secondary metabolism, and virulence" addresses the research problem in the introduction. The materials and methods section provides a detailed presentation of the research stages. The authors present the results logically and meticulously discuss them. The discussion and conclusions correspond to the research objectives, offering a thorough analysis tied to other research outcomes.

The manuscript effectively introduces and delineates the research problem, establishing a comprehensive foundation for the study. The materials and methods section systematically details the procedural aspects, enhancing the methodological transparency of the study. The coherent and logical presentation of results, coupled with an in-depth discussion, facilitates a nuanced understanding of the findings. Importantly, the discussion and conclusions aptly relate to the research's purpose and findings, drawing connections to existing research outcomes.

The manuscript is well written, however, it requires minor editorial adjustments. Specifically, the italicization of the term "in vivo" in line 68 and italicization of Latin names such as A. fumigotus and Aspergillus in lines 278, 279, 280, and subsequent lines should be addressed. Additionally, in Figure 4, it is advised to maintain uniformity by using the same scale from 5 to -15 for E, F, G to ensure consistency across the illustration.

Overall, the manuscript presents an interesting and important research topic, showcasing a well-structured narrative with a robust methodology and insightful discussions. The minor editorial adjustments will further enhance the manuscript's clarity and precision.

Author Response

***Thank you very much for your comments.  It seems  that  italics were lost during submission.  We have fixed this problem in the following locations in the revised manuscript.  These are also indicated in the “Track changes” copy of this document, indicated in blue font.

Line   68: in vivo

          281: Aspergillus fischeri

          282: Aspergillus flavus

          283: A. nidulans

          285: Aspergillus

          291: Saccharomyces pombe

          311: pyrG 

          329: A. nidulans

          335, 351: brlA, abaA, wetA

          327, 349, 354, 430, 460, 483, 484, 487: P

          407: Aspergillus fumigatus 

          427: fmqD, pdsA, pypC, and fumR

          438: fmqD

          440: pdsA

          441: pypC

          442: fumR

          445: psoA

          450: gliZ

          451: gliP

          453, 456, 458: veA, laeA

          278, 280, 284, 286, 287, 309, 323, 329, 331, 349, 356, 358, 364, 368, 378, 391, 399, 423, 456, 475, 476 : A. fumigatus 

          309, 311, 312, 313, 316, 318, 319, 320, 323, 330, 331, 332, 334, 336, 337, 340, 343, 349, 350, 351, 356, 357, 364, 370, 372, 376, 378, 383, 385, 387, 391, 396, 402, 403, 404, 409, 411, 415, 417, 422, 423, 436, 437, 438, 444, 445, 447, 448, 450, 452, 453, 456, 457, 463, 464, 465, 467, 470, 471, 475, 476, 483, 487 : osaA

          279, 285: Aspergillus

***We have also made modifications in Figure 4, graphs E,F,G, changing the Y-axis scale and made it 5 to -20 for consistency across the graphs.